# Modeling the Kinetics of Polyethylene Terephthalate and Polyesters with Terminal Hydroxyl Groups Transesterification Reactions

**DOI:** 10.3390/polym17070992

**Published:** 2025-04-06

**Authors:** Kirill A. Kirshanov, Roman V. Toms, Gadir Sh. Aliev, Daniil A. Ismaylov, Natalya Yu. Shagina, Pavel V. Sokolovskiy, Guliya R. Nizameeva, Alexander Yu. Gervald

**Affiliations:** 1M.V. Lomonosov Institute of Fine Chemical Technologies, MIREA—Russian Technological University, 119571 Moscow, Russia; zaligad99@gmail.com (G.S.A.); ismaylov.daniil@bk.ru (D.A.I.); gervald@bk.ru (A.Y.G.); 2Department of Oncology, I.M. Sechenov First Moscow State Medical University, 119435 Moscow, Russia; shagina_n_yu@staff.sechenov.ru; 3N.D. Zelinsky Institute of Organic Chemistry, Russian Academy of Sciences, 119991 Moscow, Russia; carbsorb@gmail.com; 4Arbuzov Institute of Organic and Physical Chemistry, FRC Kazan Scientific Center, Russian Academy of Sciences, 420088 Kazan, Russia; guliya.riv@gmail.com; 5Department of Physics, Kazan National Research Technological University, 68, K. Marx Str., 420015 Kazan, Russia

**Keywords:** polyethylene terephthalate, PET, polyester, reactive oligomers, polymer waste, chemical recycling, transesterification, interchain exchange, ester exchange

## Abstract

Interchain exchange, proceeded by the transesterification mechanism, allows one to obtain polyethylene terephthalate-based polyester products, bypassing the stage of molecular weight reduction and polycondensation used in classical methods of chemical recycling. A kinetic model is presented, which describes the change in the concentrations of bound and terminal units of ethylene glycol from PET and glycol from another polyester, as well as free molecules of ethylene glycol and another glycol, during transesterification reactions for the first time. Experimental data on the dependence of the degree of randomness and conversion on timeduring the interaction of polyethylene terephthalate and oligodiethylene terephthalate with terminal hydroxyl groups with a number-average molecular weight of 860 g/mol in different ratios were obtained. Molecular weight characteristics of the products of PET and oligoesters with hydroxyl end group interchain exchange, with number-average molecular weights from 610 to 860 g/mol, were also investigated. The simulation results were also compared with published data on the dependence of the degree of randomness and conversion on time during ether exchange in PET/PEN blends. The developed kinetic model was found to be in agreement with the experimental data.

## 1. Introduction

Polyethylene terephthalate (PET) is one of the most widely produced polymers. High volumes of PET production make the problem of its waste disposal relevant.

The main ways to manage polyethylene terephthalate waste are recycling, landfilling, and incineration [1,2]. The most preferable method from the point of view of environmental protection is recycling. Currently, the main focus is physical recycling, sometimes also called mechanical recycling [3]. Chemical recycling methods also have high potential. In particular, they make it possible to recycle contaminated waste, including medical waste. Depending on the chemical composition and molecular weight characteristics of the agents, the products of chemical recycling can be either monomers for the synthesis of polyesters [4,5,6,7] or various copolyesters [8,9,10,11,12,13], including polyester resins [14,15].

Chemical recycling of polyethylene terephthalate is classified depending on the nature of the reactive groups in the agent: hydrolysis, alcoholysis, glycolysis, aminolysis, ammonolysis and other methods [16,17]. Alcoholysis, glycolysis (as a special case of alcoholysis), acidolysis and ester exchange are based on transesterification reactions [3]. Also, during the reactions, there may be an increase or decrease in molecular weight, in which case the processes are referred to as polycondensation or degradation, respectively. If the process changes the molecular weight distribution and averages the monomer composition of PET and other polyester, the process is called interchain exchange [3]. In polyethylene terephthalate recycling, the original PET waste has a high molecular weight. If the intended product is also a polymer or oligomer with a high molecular weight, then obtaining a low-molecular semi-finished product is not an effective process. It has been previously shown [7,14] that interchain exchange processes are more effective than degradation.

Among the interchange reactions in polyester/polyester blends [18], the most researched system is PET/PEN. There are also a number of works devoted to the mixture of PET and PBT and the exchange of PET with bisphenol-A polycarbonate [19,20].

The rate of interchange reactions of PET with other polyesters is described in different ways, both using chemical kinetics and using mathematical approaches such as the Monte Carlo method [21] and the RSM approach [22]. Most often, the kinetics of the exchange reactions of PET and PEN are described by the general Equation (1) [23,24]:(1)A+B⇄2C,
where A—PET, B—PEN, and C—transesterification products.

This equation is equivalent to the chemical reaction (2):(2)TET+NEN⇄2TEN,
where T—terephthalate unit, N—naphthalate unit, E—ethylene glycol unit.

In [23], the kinetic description of reaction (1) was used to estimate the critical transesterification temperature during the extrusion of a PET/PEN blend. A similar kinetic model was used in [24]. The authors determined the frequency factors and activation energies by approximating the model to experimental data. The proposed model shows high agreement with experiment. However, the obtained rate constants differ by several orders of magnitude from those determined for ester exchange in PET/PEN mixtures [21] based on studies of reactions between hydrogenous and deuterated poly(ethylene terephthalate) [25].

It is known that terminal hydroxyl groups have a significant effect on the rate of ester interchange reactions [26]. A significant effect of terminal groups and residual catalysts on the rate of interchain exchange reactions is also noted by Krentsel et al. [27]. Terminal hydroxyl groups are the most active groups in transesterification reactions.

The description of the kinetics of reactions in polyethylene terephthalate involving terminal hydroxyl groups is well known. To obtain high-molecular polyester, a process is carried out in which the end hydroxyl groups of bis(2-hydroxyethyl) terephthalate and the resulting oligoethylene terephthalates react with the ester groups of BHET and oligoethylene terephthalates [28,29]. To describe the kinetics of this process, an approach is used that takes into account the change in the concentration of terminal ethylene glycol units ${EG}_{t}$, bound ethylene glycol units in the chain ${EG}_{b}$, and free ethylene glycol EG (3):(3)$${EG}_{t}+{EG}_{t}⇄{EG}_{b}+EG$$

In this case, the influence of the length of the chains to which the terminal or linked unit of ethylene glycol is attached is not taken into account.

In works devoted to the kinetics of chemical recycling of polyethylene terephthalate—in particular, the kinetics of glycolysis under the action of ethylene glycol—a similar approach is used. Thus, Javed et al. [30] take into account only the reaction of formation of bis(2-hydroxyethyl) terephthalate, which, in the annotation used here, corresponds to the following reaction (4):(4)$${EG}_{b}+EG\to {EG}_{t}+{EG}_{t}$$

The described approach does not take into account the reactions of hydroxyl groups in the terminal ethylene glycol units with the ester groups of the bound ethylene glycol units (5):(5)$${EG}_{t}+{EG}_{b}⇄{EG}_{b}+{EG}_{t}$$

When describing polycondensation kinetics, taking this process into account is meaningless, since the reactants and products of this reaction are identical. However, in the case of glycolysis, the reaction of the glycolysis agent hydroxyl group with the initial polyester ester group will in any case lead to a more rapid decrease in molecular weight. In a previous paper [31], we proposed an approach that allows us to take such reactions into account. For this purpose, all ethylene glycol units in the reaction mixture were divided into those present in the initial polyester and the glycolysis agent. In this case, the formation of terminal groups and free glycols in situ was taken into account. The process is described by 12 reactions.

The aim of this work is to develop and verify a kinetic model of transesterification reactions between polyethylene terephthalate and polyesters with terminal hydroxyl groups. The model being developed must take into account the formation and reactivity of terminal hydroxyl groups during the process.

## 2. Materials and Methods

### 2.1. Materials

Polyethylene terephthalate was used in the form of PET flakes (KIREFIR LLC, Moscow, Russia). The monomers dimethyl terephthalate, phthalic anhydride, ethylene glycol, diethylene glycol and 1,2-propylene glycol, as well as the catalyst antimony trioxide, were purchased from Sigma Aldrich (St. Louis, MO, USA) and were used without further purification.

### 2.2. Synthesis of Oligoesters with Terminal Hydroxyl Groups

To study the influence of terminal hydroxyl groups, polyesters with a high concentration of these groups and, consequently, low molecular weights were obtained. Therefore, from here on, these polymers are called oligoesters. Oligodiethylene terephthalate (ODET), oligoethylene phthalate (OEP), oligopropylene terephthalate (OPT), and oligopropylene phthalate (OPP) samples were synthesized as follows: phthalic monomer (dimethyl terephthalate or phthalic anhydride) and glycol (ethylene glycol, diethylene glycol or 1,2-propylene glycol) were mixed in a molar ratio of 4:5, respectively. Antimony trioxide catalyst was used in an amount of 0.1 wt%. Initially, the monomer mixture was kept at a temperature of 140 °C with stirring at 100 rpm for 90 min. After that, the temperature was gradually increased by 10 °C every 30 min, and the stirring speed by 50 rpm until 190 °C and 350 rpm were reached. The reaction was then carried out until the end of the isolation of low-molecular compounds under a vacuum of 40 mbar.

### 2.3. PET and Oligoesters with Terminal Hydroxyl Groups Interchain Exchange

The transesterification reactions of PET and oligoesters with terminal hydroxyl groups were carried out as follows. First, PET was melted at 280 °C. The melting of PET was followed by the introduction of ODET, OEP, OPET, or OPP oligoester. The transesterification catalyst is antimony trioxide, which is present in the previously prepared oligoesters. The reaction was carried out for 90 min with stirring at 50 rpm, with samples being taken after 7.5, 15, 30, 60, and 90 min. The transesterification products are oligo(diethylene-co-ethylene terephthalate) (ODEET), oligo(ethylene phthalate-co-terephthalate) (OEPT), oligo(propylene-co-ethylene terephthalate) (OPET), and oligo(propylene-co-ethylene phthalate-co-terephthalate) (OPEPT), respectively.

### 2.4. Characterization of PET, Oligoesters and Transesterification Products

#### 2.4.1. Fourier Transform Infrared (FTIR) Spectroscopy

Fourier transform infrared (FTIR) spectroscopy was used to confirm the monomer composition of the polyesters [14]. A Spectrum 65 FT-IR spectrometer (Perkin Elmer, Waltham, MA, USA) was used.

#### 2.4.2. Viscosimetry

The degree of polycondensation of polyethylene terephthalate used to calculate the input data for the kinetic modeling was determined based on the viscosity-average molecular weight, measured by viscometry, according to ASTM D4603 [32]. The ranges of the Mark–Houwink constants were taken to be from 3.72 × 10^−4^ to 4.68 × 10^−4^ for K and from 0.68 to 0.73 for a.

#### 2.4.3. Differential Scanning Calorimetry (DSC)

Thermal properties (glass transition temperature, melting point, degree of crystallinity) of the polyethylene terephthalate waste were determined by differential scanning calorimetry (DSC) [33]. A DSC 204 F1 Phoenix calorimeter (NETZSCH Geratebau GmbH, Selb, Germany) was used.

#### 2.4.4. Gel Permeation Chromatography (GPC)

The average number and average molecular weights of the transesterification products were determined by gel permeation chromatography (GPC). A Gilson chromatograph (Gilson Inc., Middleton, WI, USA) with Agilent MIXED-E column (Agilent, Santa Clara, CA, USA) was used.

#### 2.4.5. Oligoester Synthesis Conversion

The oligoester synthesis conversion α, %, was calculated by adjusting the parameter in Equation (8) according to the previously described method [14].

The Gardner scale was used to determine color according to ASTM D1544 [34].

#### 2.4.6. Nuclear Magnetic Resonance (NMR) Spectroscopy

NMR spectroscopy is the most widely used method for studying the composition of ethylene terephthalate-containing copolymers [35]. The spectra were obtained by means of ^1^H NMR spectrometer BrukerDPX-500 (Bruker Corporation, Ettlingen, Germany) using CDCl_3_ as a solvent. The spectra were processed using NMRium software (v. 0.60.0).

The conversion of transesterification reactions X, %, and the degree of randomness RD were determined from the area of peaks in the NMR spectra corresponding to different sequences [21,24] contained in PET (xx), oligoesters (yy), and their reaction products (xy). The peak areas are equal to $A_{xx}$, $A_{yy}$, and $A_{xy}$, respectively.

The total area under the peaks is determined using Equation (6):(6)$$A=A_{xx}+A_{xy}+A_{yy}$$

The mole fractions of the sequences xx, xy and yy are determined according to Equations (7)–(9):(7)$$P_{xx}=\frac{A_{xx}}{A}$$(8)$$P_{xy}=\frac{A_{xy}}{A}$$(9)$$P_{yy}=\frac{A_{yy}}{A}$$

Therefore, the mole fractions of the x and y units are determined using Equations (10)–(12):(10)$$P_{x}=P_{xx}+\frac{P_{xy}}{2}$$(11)$$P_{y}=P_{yy}+\frac{P_{xy}}{2}$$(12)$$P_{x}+P_{y}=1$$

Thus, the degree of randomness and conversion along the links x, X(x), and y, Y(y), are determined using Equations (13)–(15):(13)$$RD=\frac{P_{xy}}{2P_{x}}+\frac{P_{xy}}{2P_{y}}$$(14)$$X\left(x\right)=\frac{P_{x}-P_{xx}}{P_{x}}\times 100\%$$(15)$$X\left(y\right)=\frac{P_{y}-P_{yy}}{P_{y}}\times 100\%$$

## 3. Results

### 3.1. PET Flakes Characterization

The composition of the PET flakes was confirmed by FTIR spectroscopy; the averaged FTIR spectrum is shown in Figure 1. The polymer waste under study is partially crystalline polyethylene terephthalate.

The molecular weight of PET flakes was studied by viscometry. The determined values of intrinsic viscosity were 0.59 dL/g, which corresponds to a viscosity-average molecular weight of 24 to 36 kg/mol and a viscosity-average polycondensation degree of 126 to 189, depending on the Mark–Houwink equation constants used. When calculating the input data for the simulation, the polycondensation degree of PET was assumed to be 150.

The melting point determined by the DSC method was 80.9 °C, the melting point was 249.8 °C, and the initial degree of crystallinity was 11.78%. According to the literature, when measured by the differential scanning calorimetry method, depending on the size and shape of the particles, the glass transition temperature of polyethylene terephthalate lies in the range of 70 to 85 °C, and the melting point is from 245 to 255 °C. The degree of crystallinity of polyethylene terephthalate can reach values of up to 50% [36]. The thermal characteristics of PET flakes fall within these ranges of values, which is expected for polyethylene terephthalate.

### 3.2. Oligoesters Characterization

FTIR spectra and GPC curves of the obtained oligoesters with terminal hydroxyl groups are shown in Figure 2.

These samples have characteristic absorption bands at 3350 cm^−1^ (hydroxyl group, stretching), 1715 cm^−1^ (carbonyl group, stretching), and 1243 cm^−1^ (ester group, stretching). The FTIR spectrum of the ODET sample has pronounced peaks at 1270 and 939 cm^−1^, which correspond to the ether group (C-O-C).

The molecular weight characteristics calculated from the results of GPC analysis are given in Table 1.

The obtained number-average molecular weights were used to calculate the input data for the simulation. Thus, the concentrations of c_6_ and c_10_ are determined by the average molecular weight (Table 1).

### 3.3. Kinetic Model Development

The kinetic model being developed is based on taking into account the interactions between the bound and terminal units of ethylene glycol from PET and glycol from other polyesters, while taking into account neighboring units of terephthalate and/or other acidic monomers, as well as free molecules of ethylene glycol and other glycol. The abbreviations used in the kinetic model are given in Table 2.

The assumptions made to simplify the model are close to those we made earlier in [31], so they are presented here in abbreviated form:Polyesters within the same source (PET or other polyester) are assumed to be in equilibrium.Antimony trioxide was used as a catalyst, with the concentration of the catalyst [c_cat_] in the reaction mixture being constant.All reactions of the same type proceeded at the same rate, regardless of the chain lengths. It is assumed that polymer chain effects, such as steric or neighbor effects, are included in the selected reaction rate constants. The pre-exponential factors and activation energies used were determined [21,28,31] to describe the kinetics of transesterification in polyethylene terephthalate in the presence of the antimony trioxide catalyst. The reaction types and their corresponding rate constants are given in Table 3.

The ester exchange rate constant can be determined from Equation (16):(16)$$k_{1}=A_{1}\cdot e^{-\frac{E_{a1}}{R\cdot T}}$$
where k_1_ is the ester exchange rate constant; A_1_ is the pre-exponential factor, 7.80·10^5^ L/(mol·min) [21]; E_a1_ is the activation energy, 105 kJ/mol [21]; R is the universal gas constant, 8.310 J/mol·K; and T is the temperature, K.

The effective polycondensation rate constant $k_{2}^{\prime }$ can be determined from Equation (17):(17)$$k_{2}^{\prime }=\left[c_{cat}\right]{\cdot k}_{2}=\left[c_{cat}\right]\cdot A_{2}\cdot e^{-\frac{E_{a2}}{R\cdot T}}$$
where A_2_ is equal to 5.66·10^8^ L^2^/(mol^2^·min) [28,31], and E_a2_ is equal to 18.5 cal/mol [28,31].

The effective alcoholysis rate constant $k_{3}^{\prime }$ can be determined from Equation (18):(18)$$k_{3}^{\prime }=k_{2}^{\prime }/K,$$
where K is the equilibrium constant, 0.5 [28,31].

4.The plug flow reactor model is used, including all assumptions.5.It is assumed that all terminal groups of polyethylene terephthalate, other polyesters and their reaction products are hydroxyl. Side reactions are not taken into account.

Table 4 shows the reactions occurring during interchain exchange between PET and another polyester (PDP). The table includes:Reactions 1–11, occurring between two bound glycols.Reactions 12–35, occurring between the terminal units of glycols with hydroxyl groups and bound glycols.Reactions 36–49, occurring between the bound units of glycols and free glycols.Reactions 50–63, occurring between two terminal units of glycols.Reactions 64–67, occurring between the terminal units of glycols and free glycols.

The reaction mechanisms are considered to be similar to those described in the previous work [31].

The material balance equations corresponding to changes in the concentrations of the reactants are presented below in the form of Equations (19)–(30):(19)$$\frac{d(c_{1})}{dt}=\;R_{8}\;+\;R_{9}\;+\;R_{13}\;+\;R_{14}\;+\;R_{16}\;+\;R_{50}\;+\;R_{51}\;-R_{1}\;-R_{2}\;-R_{3}\;-R_{18}\;-\;R_{24}\;-R_{30}\;-\;R_{36}\;-\;R_{43}$$(20)$$\frac{d(c_{2})}{dt}=R_{4}\;+\;R_{9}\;+\;R_{10}\;+\;R_{25}\;+\;R_{27}\;+\;R_{29}\;+\;R_{59}\;+\;R_{60}\;-R_{3}\;-R_{6}\;-R_{7}\;-R_{12}\;-R_{19}\;-R_{31}\;-R_{37}\;-\;R_{44}$$(21)$$\begin{array}{c} \frac{d\left(c_{3}\right)}{dt}=\;R_{1}\;+\;R_{2}\;+\;2R_{3}\;+\;R_{6}\;+\;R_{7}\;+\;R_{12}\;+\;R_{15}\;+\;R_{17}\;+\;R_{24}\;+\;R_{26}\;+\;R_{28}\;+\;R_{52}\;+\;R_{53}\;+\;R_{58}\\ -R_{4}\;-R_{8}\;-\;2R_{9}\;-\;R_{10}\;-\;R_{13}\;-\;R_{20}\;-R_{21}\;-\;R_{25}\;-\;R_{32}\;-\;R_{33}\;-\;R_{38}\;-\;R_{45}\;-\;R_{46} \end{array}$$(22)$$\begin{array}{c} \frac{d\left(c_{4}\right)}{dt}=\;R_{1}\;+\;R_{4}\;+\;2R_{5}\;+\;R_{6}\;+\;R_{8}\;+\;R_{19}\;+\;R_{25}\;+\;R_{22}+R_{30}\;+\;R_{33}\;+\;R_{34}\;+\;R_{56}\;+\;R_{57}\;+\;R_{61}\\ -R_{2}\;-\;R_{7}\;-\;R_{10}\;-\;2R_{11}\;-R_{16}\;-\;R_{17}\;-\;R_{23}\;-\;R_{28}\;-\;R_{29}\;-\;R_{35}\;-\;R_{41}\;-\;R_{42}\;-\;R_{49} \end{array}$$(23)$$\frac{d(c_{5})}{dt}=\;R_{2}\;+\;R_{10}\;+\;R_{11}\;+\;R_{18}\;+\;R_{20}\;+\;R_{23}\;+\;R_{54}\;+\;R_{55}\;-\;R_{5}\;-\;R_{6}\;-\;R_{8}\;-\;R_{14}\;-\;R_{26}\;-\;R_{34}\;-R_{39}\;-\;R_{47}$$(24)$$\frac{d(c_{6})}{dt}=\;R_{7}\;+\;R_{11}\;+\;R_{31}\;+\;R_{32}\;+\;R_{35}\;+\;R_{62}\;+\;R_{63}\;-\;R_{1}\;-\;R_{4}\;-\;R_{5}\;-\;R_{15}\;-\;R_{22}\;-\;R_{27}\;-\;R_{40}\;-\;R_{48}$$(25)$$\begin{array}{c} \frac{d\left(c_{7}\right)}{dt}=R_{18}\;+R_{21}\;+R_{24}\;+R_{25}\;+R_{30}\;+\;R_{32}\;+2R_{36}\;+R_{38}\;+\;R_{39}\;+\;R_{41}\;+R_{43}\;+R_{45}\;+\;R_{65}\\ \;-R_{12}\;-R_{13}\;-R_{14}\;-R_{15}\;-\;R_{16}\;-R_{17}\;-2R_{50}\;-R_{51}\;-R_{52}\;-R_{53}\;-R_{54}\;-R_{61}\;-R_{64} \end{array}$$(26)$$\begin{array}{c} \frac{d(c_{8})}{dt}=R_{14}+R_{17}+R_{26}+R_{29}+R_{34}+R_{35}+R_{39}+R_{42}+R_{43}+R_{46}+2R_{47}+R_{49}+R_{64}\\ \;-R_{18}-R_{19}-R_{20}-R_{21}-R_{22}-R_{23}-R_{51}-R_{54}-2R_{55}-R_{56}-R_{57}-R_{58}-R_{65} \end{array}$$(27)$$\begin{array}{c} \frac{d(c_{9})}{dt}=R_{12}+R_{13}+R_{19}+R_{20}+R_{31}+R_{33}+2R_{37}+R_{38}+R_{40}+R_{42}+R_{44}+R_{46}+R_{67}\\ \;-R_{24}-R_{25}-R_{26}-R_{27}-R_{28}-R_{29}-R_{52}-R_{56}-R_{58}-2R_{59}-R_{60}-R_{62}-R_{66} \end{array}$$(28)$$\begin{array}{c} \frac{d(c_{10})}{dt}=\;R_{15}\;+\;R_{16}\;+\;R_{22}\;+\;R_{23}\;+\;R_{27}\;+\;R_{28}\;+\;R_{40}\;+\;R_{41}\;+\;R_{44}\;+\;R_{45}\;+\;2R_{48}\;+\;R_{49}\;+\;R_{66}\\ \;-R_{30}\;-\;R_{31}\;-R_{32}\;-R_{33}\;-R_{34}\;-\;R_{35}\;-R_{53}\;-\;R_{57}\;-\;R_{60}\;-\;R_{61}\;-\;R_{62}\;-\;2R_{63}\;-\;R_{67} \end{array}$$(29)$$\frac{d(c_{11})}{dt}=\;R_{50}\;+\;R_{52}\;+\;R_{54}\;+\;R_{56}\;+\;R_{59}\;+\;R_{61}\;+\;R_{62}\;+\;R_{64}\;+\;R_{66}\;-\;R_{36}\;-\;R_{37}\;-\;R_{38}\;-\;R_{39}\;-\;R_{40}\;-\;R_{41}\;-\;R_{42}\;-\;R_{65}\;-\;R_{67}$$(30)$$\frac{d(c_{12})}{dt}=R_{51}\;+R_{53}\;+\;R_{55}\;+\;R_{57}\;+\;R_{58}\;+\;R_{60}\;+R_{63}+R_{65}\;+R_{67}\;-R_{43}\;-R_{44}\;-R_{45}\;-\;R_{46}\;-R_{47}\;-R_{48}\;-R_{49}\;-\;R_{64}\;-R_{66}$$

The resulting system of differential equations can be solved using numerical methods.

### 3.4. Degree of Randomness and Conversion Model Calculations

To calculate the degree of randomness and conversion from the model data, Equations (13)–(18) can be used. In this case, the concentrations of units and sequences for substitution into these equations can be calculated using the concentrations c_1_–c_12_—the output data of the model. In this case, three different cases are possible:E ≠ D, T = P.E = D, T ≠ P.E ≠ D, T ≠ P.

If E = D and T = P, the model can be simplified to that described in [31].

In the first case, interchain exchange occurs between polyethylene terephthalate and another polyether terephthalate. Then, Equations (31)–(33) are valid:(31)$$P(ED)=\frac{\begin{array}{c} 4c_{1}\cdot c_{5}+2c_{1}\cdot c_{4}+4c_{6}\cdot c_{2}+2c_{6}\cdot c_{3}+2c_{2}\cdot c_{4}+2c_{5}\cdot c_{3}+2c_{3}\cdot c_{4}+2c_{7}\cdot c_{5}\\ +c_{7}\cdot c_{4}+2c_{8}\cdot c_{1}+c_{8}\cdot c_{3}+2c_{9}\cdot c_{6}+c_{9}\cdot c_{4}+2+c_{10}\cdot c_{2}+c_{10}\cdot c_{3}+c_{7}\cdot c_{8}+c_{9}\cdot c_{10} \end{array}}{\sum_{i=1}^{10}c_{i}}$$(32)$$P\left(E\right)=c_{1}+c_{2}+c_{3}+c_{7}+c_{9}$$(33)$$P(D)=c_{4}+c_{5}+c_{6}+c_{8}+c_{10}$$

The second case describes, for example, the classical PET/PEN system. For this case, Equations (34)–(36) are valid:(34)$$P\left(TP\right)=c_{3}+c_{4}$$(35)$$P\left(T\right)=c_{1}+0.5c_{3}+0.5c_{4}+c_{5}+0.5c_{7}+0.5c_{8}$$(36)$$P(P)=c_{6}+0.5c_{4}+0.5c_{3}+c_{5}+0.5c_{9}+0.5c_{10}$$

It is noteworthy that P(T) and P(P) include not only 0.5P(TP), equal to $0.5c_{3}+0.5c_{4}$, but also $0.5c_{7}+0.5c_{8}$ and $0.5c_{9}+0.5c_{10}$, respectively, taking into account the contribution of the terminal links to the concentration of T and P links. At high molecular weights of polymers, this term can be neglected.

In the third case, P(E), P(D), P(T), and (P(P) can be determined according to Equations (32), (33), (35), and (36). Using the same assumptions as made by Devaux et al. [19], the degree of randomness and conversion can be calculated through P(EP) and P(DT), calculated according to Equations (37) and (38):(37)$$P\left(EP\right)=2c_{2}+c_{3}+c_{9}$$(38)$$P\left(DT\right)=2c_{5}+c_{4}+c_{8}$$

### 3.5. Molecular Weight Characteristics Model Calculations

The molecular weight characteristics were calculated according to the model based on the concentrations of the reactants. For comparison with the experimental results, the molecular weight distribution was estimated according to the Flory–Schulz equation, as described earlier in [31].

Minor changes relative to [31] have been made to the equations for calculating the degrees of polycondensation. The average degrees of polycondensation can be determined from the ratio of terminal and bonded glycol units according to Equations (39)–(41):(39)$${\bar{X}}_{t}^{h}=\frac{c_{1t}+c_{2t}+c_{3t}+0.5(c_{7t}+c_{9t})}{0.5(c_{7t}+c_{9t})}$$
where ${\bar{X}}_{t}^{h}$ is the average degree of PET over time t, and $c_{it}$ is concentration c_i_ over time t.(40)$${\bar{X}}_{t}^{l}=\frac{c_{4t}+c_{5t}+c_{6t}+0.5(c_{8t}+c_{10t})}{0.5(c_{8t}+c_{10t})}$$
where ${\bar{X}}_{t}^{l}$ is the average degree of PDP oligoester over time t.(41)$${\bar{X}}_{t}^{o}=\frac{c_{1t}+c_{2t}+c_{3t}+c_{4t}+c_{5t}+c_{6t}+0.5(c_{7t}+c_{8t}+c_{9t}+c_{10t})}{0.5(c_{7t}+c_{8t}+c_{9t}+c_{10t})}$$
where ${\bar{X}}_{t}^{o}$ is the overall average degree of polycondensation of polyester over time t.

### 3.6. Conversion and Degree of Randomness in PET/ODET Blends

Figure 3 shows the HNMR spectra of the different products obtained.

In the spectra corresponding to the OPEPT and OPET polymers, a set of inseparable signals is present in the region of 8.0–8.1 ppm. They presumably correspond to the chemical shift in the protons of terephthalates located between two ethylene glycol units, two 1,2-propylene glycol units, or an ethylene glycol unit and a 1,2-propylene glycol unit, taking into account the various options for the addition of 1,2-propylene glycol to the terephthalate.

The characteristic signals in the spectrum of the OPET sample should lie in the region of 5.5–6.0 ppm [37], however no fine splitting was observed.

For the ODEET sample in the region of 8.00–8.15 ppm there are three distinguishable signals, shown in Figure 4. They correspond to the triads ETE, ETD, and DTD [38]. For verification of the model, the interchain exchange of PET and ODEET polyester was selected.

In all the following calculations, a step of 0.1 min was used for the numerical solution of the system of differential equations of the material balance. The temperature conditions were set as equal to the experimental ones, and the concentration of the catalyst in terms of dry Sb_2_O_3_ was set as equal to 0.0048 mol/L, which corresponds to 0.1 wt.% of the reaction mixture.

The initial concentrations used in the calculation are shown in Table 5.

The peak area data shown in Figure 4 for the studied samples at reaction times of 7.5, 15, 30, and 60 min are given in Appendix A. In Figure 5, the points corresponding to the obtained experimental data are superimposed on the curves corresponding to the simulation results.

The simulation results and experimental data are highly consistent. The calculations and average approximation errors are presented in Appendix A. The randomness degree values have an average approximation error of no more than 5%, and for conversion, it is no more than 10%. At the same time, the deviation increases with increasing concentration of ODET polyester in the initial mixture. This can be explained by the limitations of Assumption 3—the use of rate constants determined for polyethylene terephthalate, which are less suitable for diethylene glycol-containing polyester.

### 3.7. Molecular Weight Characteristics of PET/Oligoester Transesterification Products

FTIR spectra and GPC curves of the interchain exchange products are shown in Figure 6. The FTIR spectra contain absorption bands corresponding to PET and OPP, OEP, OPT, and ODEET, respectively.

The molecular weight characteristics calculated from the GPC analysis are shown in Table 2. The input concentrations used in the calculation are shown in Table 6.

By the first point for which the molecular mass distribution is experimentally determined (15 min), the interchain exchange, according to the kinetic model, reaches equilibrium. Therefore, only one simulation result is given for each sample in Table 7.

For all values of the number-average molecular weight, except for the ODEET-90 point, the deviation of the simulation result from the experimental data does not exceed 400 g/mol. As in the previous work [31], the simulated weight average molecular weights and polydispersity indices are lower than those determined experimentally. This can be explained by the limitations of using the Flory–Schulz equation to predict the molecular weight distribution.

Also, with increasing reaction time, an increase in molecular weights can be observed in the OEPT and ODEET samples, i.e., those that do not contain 1,2-propylene glycol. The increase in molecular weight can be explained by the occurrence of polycondensation with distillation of ethylene glycol, the formation of which occurs during the process at 280 °C, which is confirmed by the small peak areas of low-molecular compounds in the sample. Another possible explanation is the occurrence of an etherification side reaction.

### 3.8. Interpretation of Reactions in PET/PEN Blends

The simulation results are also compared with published data on the dependence of the degree of randomness and conversion on time during interchain exchange in PET/PEN mixtures [21,24]. The temperatures of 280 and 300 °C indicated by the authors were used in the simulation. Since the authors did not specify the nature and concentration of the catalyst (or its residues) used by them, the same value of 0.0048 mol/L was adopted in this simulation. The concentrations of the terminal units c_7_ and c_10_ were calculated based on the molecular weights of PET and PEN indicated by the authors. The input concentrations used in the calculation are shown in Table 8.

In Figure 7 and Figure 8, the points corresponding to the obtained experimental data are superimposed on the curves corresponding to the simulation results at 280 and 300 °C, respectively.

Despite a number of assumptions made associated with the use of published data, the agreement of experimental data [21,24] and the degrees of randomness and conversions calculated using the kinetic model is high. As in Figure 6, the accuracy of the model decreases with decreasing concentrations of polyethylene terephthalate in the reaction mixture.

## 4. Conclusions

Thus, a kinetic model of transesterification reactions between polyethylene terephthalate and polyesters with terminal hydroxyl groups is proposed. The model includes 67 reactions and 12 balance equations describing the change in the concentrations of bound and terminal units of ethylene glycol from PET and glycol from another polyester, taking into account the surrounding units, as well as the concentrations of free ethylene glycol and another glycol. A method for calculating the degree of randomness and conversion using the calculated concentrations is also proposed for the model.

To verify the model, experimental data on the change in the degree of randomness and conversion during interchain exchange of polyethylene terephthalate and oligodiethylene glycol terephthalate with hydroxyl end groups at a temperature of 280 °C were compared with the simulation results. The molecular weight characteristics of the reaction products of PET and various oligoesters with hydroxyl end groups were also considered. The model was found to be in good agreement with the experimental data. At the same time, a systematic deviation of the simulated weight-average molecular weights and polydispersity indices from the experimental ones was discovered. Deviations may be due to the assumptions underlying the model.

The simulation results are also compared with experimental data on ether exchange in PET/PEN blends with low end group concentrations at temperatures of 280 and 300 °C, as published by other authors.

Thus, the model can be used for the design of experiments and optimization of conditions for the ester exchange between PET and other polyesters. The proposed model can accelerate the development of technologies that allow one to obtain a wide range of polyester products based on polyethylene terephthalate.

## Figures and Tables

**Figure 1 polymers-17-00992-f001:**
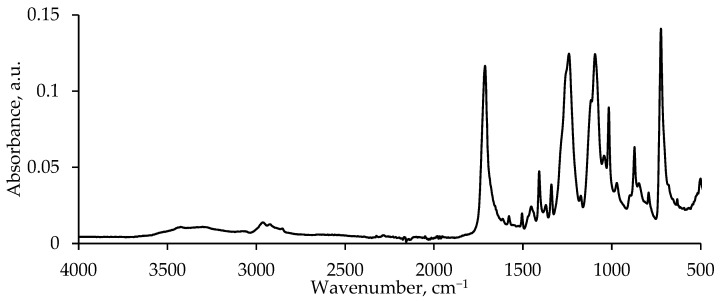
FTIR spectrum of PET flakes.

**Figure 2 polymers-17-00992-f002:**
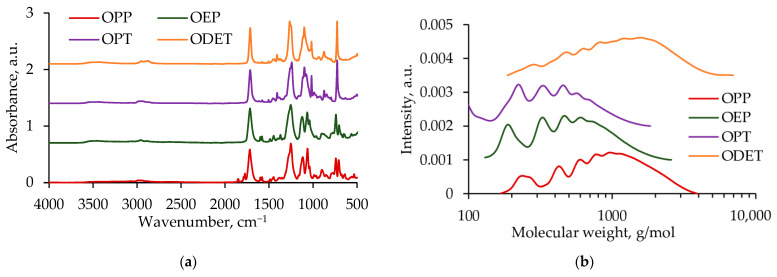
FTIR spectra (**a**) and GPC curves (**b**) of OPP, OEP, OPT, and ODET samples.

**Figure 3 polymers-17-00992-f003:**
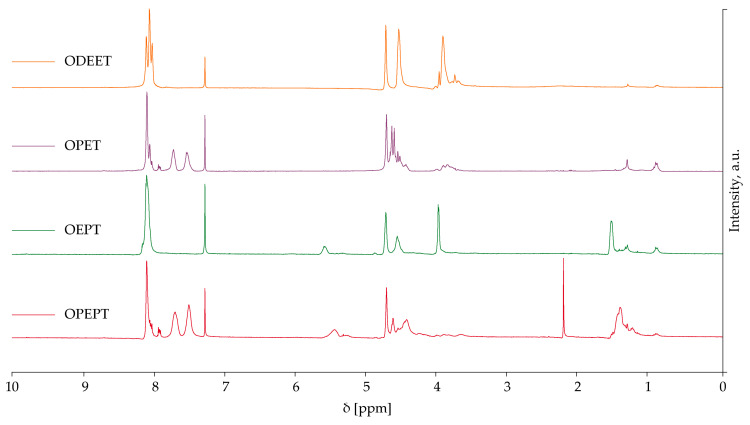
^1^HNMR spectra of OPEPT, OEPT, OPET, and ODEET products obtained in a PET:oligoester mass ratio of 50:50, reaction time 30 min.

**Figure 4 polymers-17-00992-f004:**
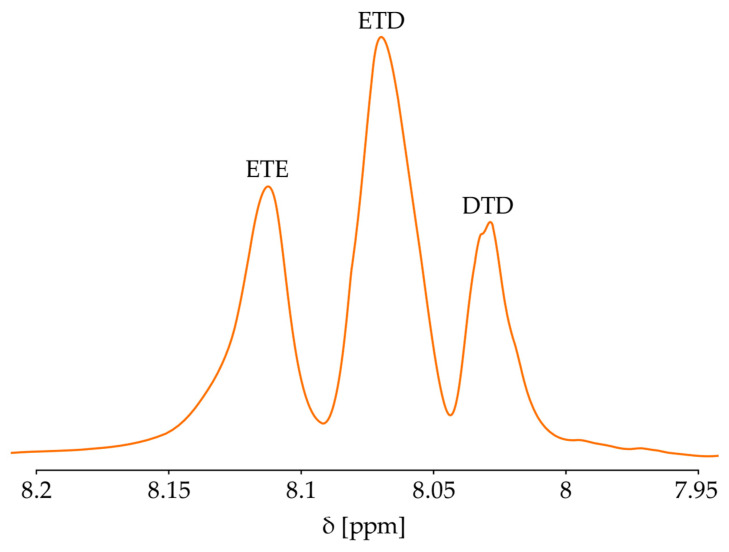
Characteristic peaks of the ETE, ETD, and DTD triads in the HNMR spectrum.

**Figure 5 polymers-17-00992-f005:**
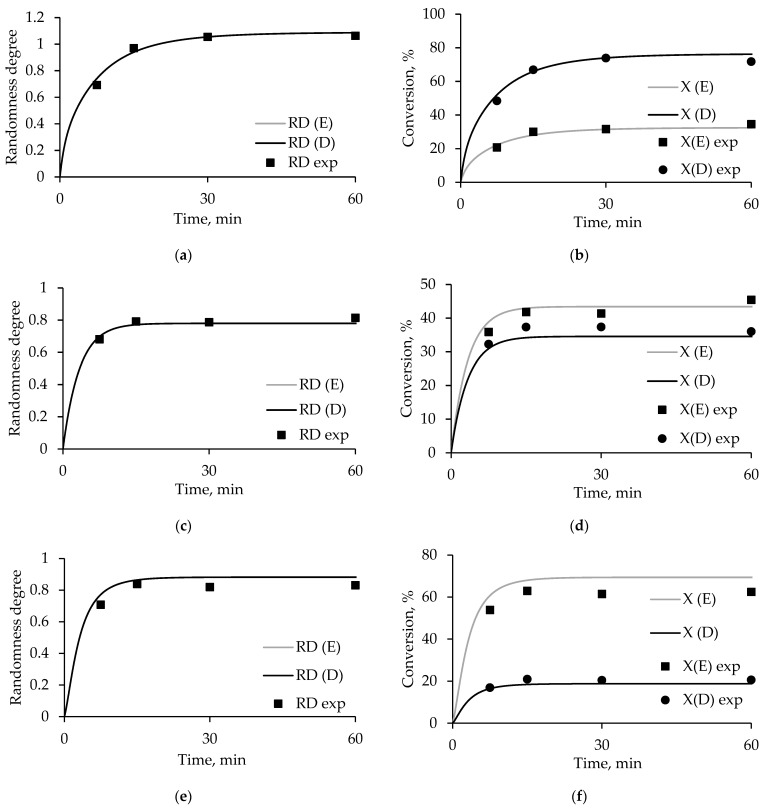
Dependence of the degree of randomness and conversion on time, where RD(E), RD(D), X(E), and X(D) are the simulation results, and RD exp and X exp are the experimental data for the products (**a**,**b**) ODEET-75:25, (**c**,**d**) ODEET-50:50, (**e**,**f**) ODEET-25:75 obtained in PET:ODET ratios of 75:25, 50:50, and 25:75, respectively.

**Figure 6 polymers-17-00992-f006:**
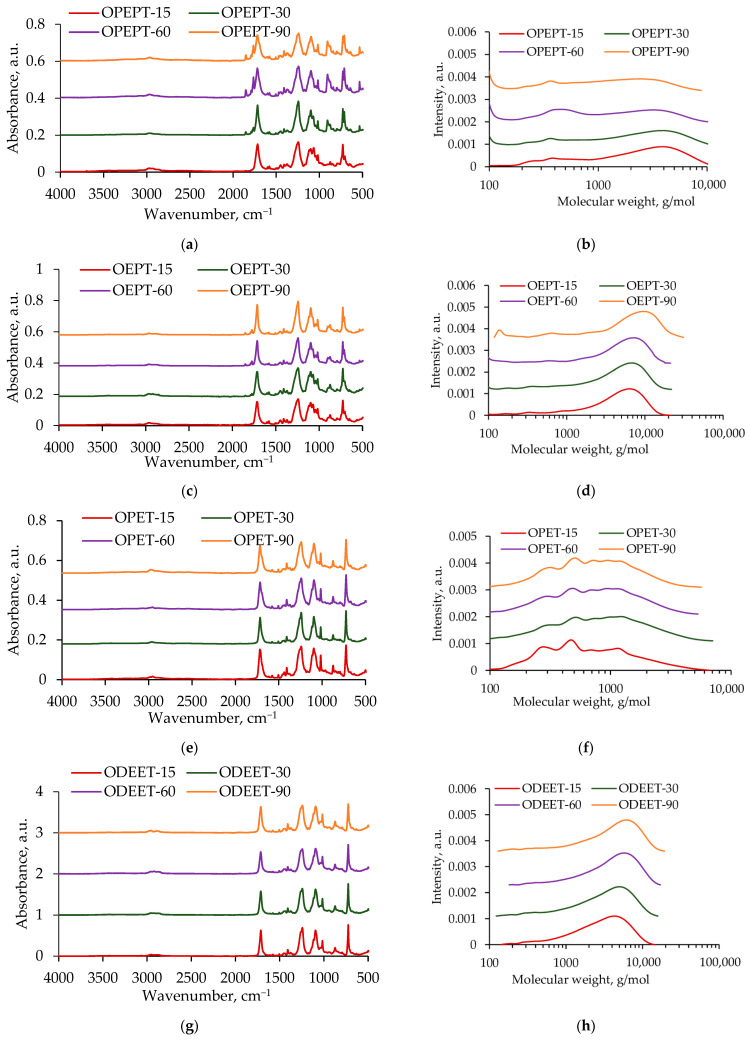
FTIR spectra and GPC curves of products (**a**,**b**) OPEPT, (**c**,**d**) OEPT, (**e**,**f**) OPET, and (**g**,**h**) ODEET; as for the mass ratio of PET, oligoester is presented in a 50:50 ratio, and the reaction time was 15, 30, 60, 90 min.

**Figure 7 polymers-17-00992-f007:**
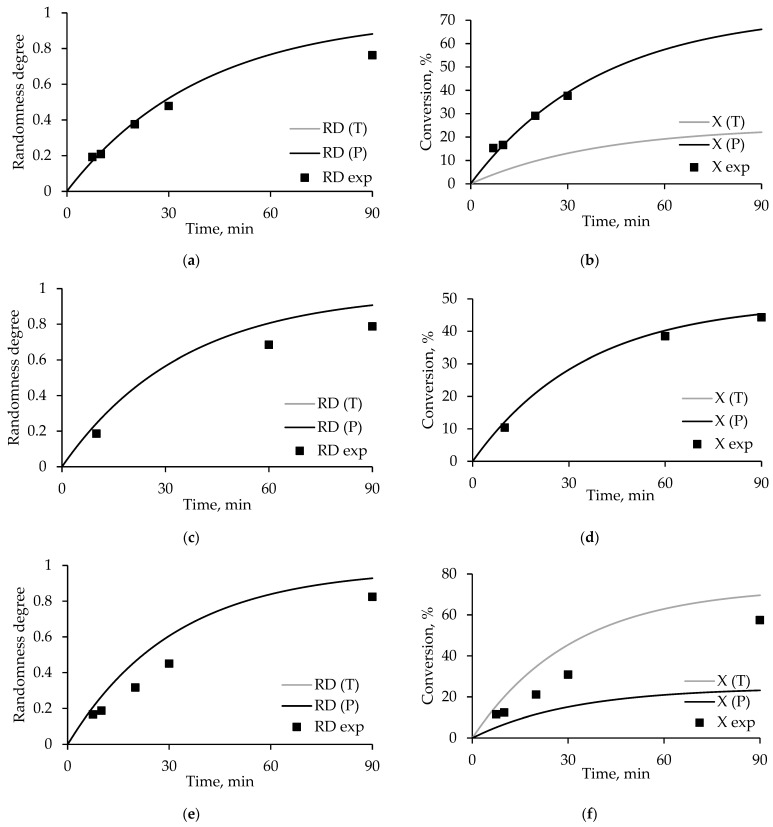
Dependence of the degree of randomness and conversion on time, where RD(E), RD(D), X(E), and X(D) are the simulation results, RD exp, X exp are the experimental data, for the products (**a**,**b**) PET/PEN-75:25, (**c**,**d**) PET/PEN-50:50, and (**e**,**f**) PET/PEN-25:75 obtained in the PET:ODET ratios of 75:25, 50:50, and 25:75, respectively, at a temperature of 280 °C.

**Figure 8 polymers-17-00992-f008:**
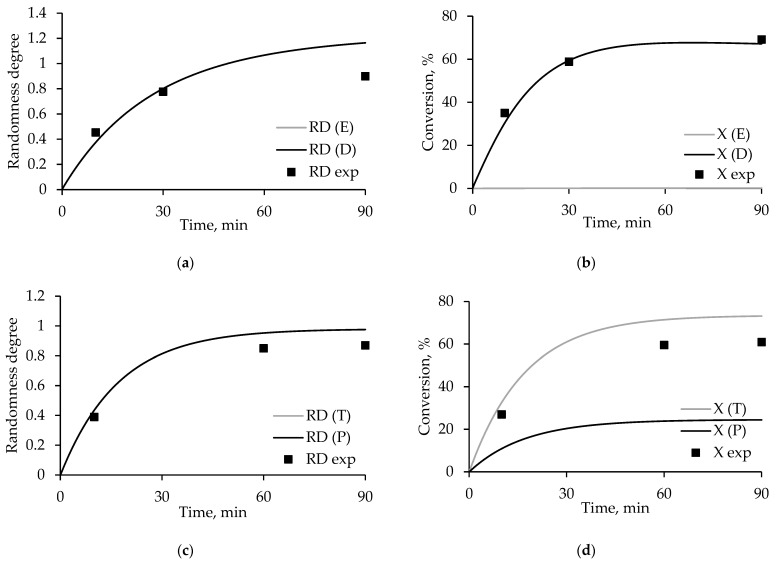
Dependence of the degree of randomness and conversion on time, where RD(E), RD(D), X(E), and X(D) are the simulation results, and RD exp and X exp are the experimental data, for the products (**a**,**b**) PET/PEN-75:25 and (**c**,**d**) PET/PEN-25:75 obtained in the PET:ODET ratios of 75:25 and 25:75, respectively, at a temperature of °C.

**Table 1 polymers-17-00992-t001:** Characteristics of OPP, OEP, OPT, and ODET samples: number average (Mn) and weight average (Mw) molecular weights, polydispersity index (PDI), conversion, and color.

Sample	Mn, g/mol	Mw, g/mol	PDI	Conversion, %	Color
OPP	667	1096	1.64	95.80	1
OEP	648	1100	1.70	96.66	3
OPT	611	932	1.53	94.34	1
ODET	857	1366	1.59	97.22	2

**Table 2 polymers-17-00992-t002:** Abbreviations used in the kinetic model are as follows: T—terephthalate unit from PET; E—ethylene glycol unit from PET; P—acidic monomer unit from polyester; D—glycol monomer unit from polyester.

Name of the Reacting Component	Abbreviation	Concentration, g/L
Bonded ethylene glycol	TET	c_1_
Bonded glycol	TDT	c_2_
Bonded ethylene glycol	TEP	c_3_
Bonded glycol	TDP	c_4_
Bonded ethylene glycol	PEP	c_5_
Bonded glycol	PDP	c_6_
Terminal ethylene glycol	TE	c_7_
Terminal glycol	TD	c_8_
Terminal ethylene glycol	PE	c_9_
Terminal glycol	PD	c_10_
Free ethylene glycol	E	c_11_
Free glycol	D	c_12_

**Table 3 polymers-17-00992-t003:** Effective rate constants of reactions.

Reaction	Reaction Rate Constant
Between two bonded glycols	$k_{1}$
Between terminal glycol and terminal or bonded glycol	$k_{2}^{\prime }$
Between free glycol and terminal or bonded glycol	$k_{3}^{\prime }$

**Table 4 polymers-17-00992-t004:** Reactions that occur during PET/polyester (PDP) interchain exchange, and the corresponding reaction rate equations.

№	Reaction	Reaction Rate Equation
1	${TET+PDP\to }^{{4k}_{1}}TEP+TDP$	$R_{1}=4k_{1}\cdot c_{1}\cdot c_{6}$
2	$TET+TDP\to^{{2k}_{1}}TEP+TDT$	$R_{2}={2k}_{1}\cdot c_{1}\cdot c_{4}$
3	$TET+PEP\to^{{4k}_{1}}2TEP$	$R_{3}={4k}_{1}\cdot c_{1}\cdot c_{2}$
4	$PDP+TEP\to^{{2k}_{1}}TDP+PEP$	$R_{4}={2k}_{1}\cdot c_{3}\cdot c_{6}$
5	$PDP+TDT\to^{{4k}_{1}}2TDP$	$R_{5}={4k}_{1}\cdot c_{5}\cdot c_{6}$
6	$PEP+TDT\to^{{4k}_{1}}TEP+TDP$	$R_{6}={4k}_{1}\cdot c_{2}\cdot c_{5}$
7	$PEP+TDP\to^{{2k}_{1}}TEP+PDP$	$R_{7}=2k_{1}\cdot c_{2}\cdot c_{4}$
8	$TDT+TEP\to^{{2k}_{1}}TDP+TET$	$R_{8}=2k_{1}\cdot c_{3}\cdot c_{5}$
9	$TEP+TEP\to^{{2k}_{1}}TET+PEP$	$R_{9}=2k_{1}\cdot c_{3}\cdot c_{3}$
10	$TEP+TDP\to^{{2k}_{1}}PEP+TDT$	$R_{10}=2k_{1}\cdot c_{3}\cdot c_{4}$
11	$TDP+TDP\to^{{2k}_{1}}TDT+PDP$	$R_{11}=2k_{1}\cdot c_{4}\cdot c_{4}$
12	$TE+PEP\to^{2k_{2}^{\prime }}TEP+PE$	$R_{12}=2k_{2}^{\prime }\cdot c_{7}\cdot c_{2}$
13	$TE+TEP\to^{k_{2}^{\prime }}TET+PE$	$R_{13}=k_{2}^{\prime }\cdot c_{7}\cdot c_{3}$
14	${TE+TDT\to }^{2k_{2}^{\prime }}TET+TD$	$R_{14}=2k_{2}^{\prime }\cdot c_{7}\cdot c_{5}$
15	${TE+PDP\to }^{2k_{2}^{\prime }}TEP+PD$	$R_{15}=2k_{2}^{\prime }\cdot c_{7}\cdot c_{6}$
16	$TE+TDP\to^{k_{2}^{\prime }}TET+PD$	$R_{16}=k_{2}^{\prime }\cdot c_{7}\cdot c_{4}$
17	$TE+TDP\to^{k_{2}^{\prime }}TEP+TD$	$R_{17}=k_{2}^{\prime }\cdot c_{7}\cdot c_{4}$
18	$TD+TET\to^{{2k}_{2}^{\prime }}TDT+TE$	$R_{18}=2k_{2}^{\prime }\cdot c_{8}\cdot c_{1}$
19	$TD+PEP\to^{{2k}_{2}^{\prime }}TDP+PE$	$R_{19}=2k_{2}^{\prime }\cdot c_{8}\cdot c_{2}$
20	$TD+TEP\to^{k_{2}^{\prime }}TDT+PE$	$R_{20}=k_{2}^{\prime }\cdot c_{8}\cdot c_{3}$
21	$TD+TEP\to^{k_{2}^{\prime }}TDP+TE$	$R_{21}=k_{2}^{\prime }\cdot c_{8}\cdot c_{3}$
22	$TD+PDP\to^{{2k}_{2}^{\prime }}TDP+PD$	$R_{22}=2k_{2}^{\prime }\cdot c_{8}\cdot c_{6}$
23	$TD+TDP\to^{k_{2}^{\prime }}TDT+PD$	$R_{23}=k_{2}^{\prime }\cdot c_{8}\cdot c_{4}$
24	$PE+TET\to^{{2k}_{2}^{\prime }}TEP+TE$	$R_{24}=2k_{2}^{\prime }\cdot c_{9}\cdot c_{1}$
25	$PE+TEP\to^{k_{2}^{\prime }}PEP+TE$	$R_{25}=k_{2}^{\prime }\cdot c_{9}\cdot c_{3}$
26	$PE+TDT\to^{{2k}_{2}^{\prime }}TEP+TD$	$R_{26}=2k_{2}^{\prime }\cdot c_{9}\cdot c_{5}$
27	$PE+PDP\to^{{2k}_{2}^{\prime }}PEP+PD$	$R_{27}=2k_{2}^{\prime }\cdot c_{9}\cdot c_{6}$
28	$PE+TDP\to^{k_{2}^{\prime }}TEP+PD$	$R_{28}=k_{2}^{\prime }\cdot c_{9}\cdot c_{4}$
29	$PE+TDP\to^{k_{2}^{\prime }}PEP+TD$	$R_{29}=k_{2}^{\prime }\cdot c_{9}\cdot c_{4}$
30	$PD+TET\to^{{2k}_{2}^{\prime }}TDP+TE$	$R_{30}=2k_{2}^{\prime }\cdot c_{10}\cdot c_{1}$
31	$PD+PEP\to^{{2k}_{2}^{\prime }}PDP+PE$	$R_{31}=2k_{2}^{\prime }\cdot c_{10}\cdot c_{2}$
32	$PD+TEP\to^{k_{2}^{\prime }}PDP+TE$	$R_{32}=k_{2}^{\prime }\cdot c_{10}\cdot c_{3}$
33	$PD+TEP\to^{k_{2}^{\prime }}TDP+PE$	$R_{33}=k_{2}^{\prime }\cdot c_{10}\cdot c_{3}$
34	$PD+TDT\to^{{2k}_{2}^{\prime }}TDP+TD$	$R_{34}=2k_{2}^{\prime }\cdot c_{10}\cdot c_{5}$
35	$PD+TDP\to^{k_{2}^{\prime }}PDP+TD$	$R_{35}=k_{2}^{\prime }\cdot c_{10}\cdot c_{4}$
36	$E+TET\to^{{2k}_{3}^{\prime }}2TE$	$R_{36}=2k_{3}^{\prime }\cdot c_{11}\cdot c_{1}$
37	$E+PEP\to^{{2k}_{3}^{\prime }}2PE$	$R_{37}=2k_{3}^{\prime }\cdot c_{11}\cdot c_{2}$
38	$E+TEP\to^{{2k}_{3}^{\prime }}TE+PE$	$R_{38}=2k_{3}^{\prime }\cdot c_{11}\cdot c_{3}$
39	$E+TDT\to^{{2k}_{3}^{\prime }}TE+TD$	$R_{39}=2k_{3}^{\prime }\cdot c_{11}\cdot c_{5}$
40	$E+PDP\to^{{2k}_{3}^{\prime }}PE+PD$	$R_{40}=2k_{3}^{\prime }\cdot c_{11}\cdot c_{6}$
41	$E+TDP\to^{k_{3}^{\prime }}TE+PD$	$R_{41}=k_{3}^{\prime }\cdot c_{11}\cdot c_{4}$
42	$E+TDP\to^{k_{3}^{\prime }}PE+TD$	$R_{42}=k_{3}^{\prime }\cdot c_{11}\cdot c_{4}$
43	$D+TET\to^{{2k}_{3}^{\prime }}TE+TD$	$R_{43}=2k_{3}^{\prime }\cdot c_{12}\cdot c_{1}$
44	$D+PEP\to^{{2k}_{3}^{\prime }}PE+PD$	$R_{44}=2k_{3}^{\prime }\cdot c_{12}\cdot c_{2}$
45	$D+TEP\to^{k_{3}^{\prime }}TE+PD$	$R_{45}=k_{3}^{\prime }\cdot c_{12}\cdot c_{3}$
46	$D+TEP\to^{k_{3}^{\prime }}PE+TD$	$R_{46}=k_{3}^{\prime }\cdot c_{12}\cdot c_{3}$
47	$D+TDT\to^{{2k}_{3}^{\prime }}2TD$	$R_{47}=2k_{3}^{\prime }\cdot c_{12}\cdot c_{5}$
48	$D+PDP\to^{{2k}_{3}^{\prime }}2PD$	$R_{48}={2k}_{3}^{\prime }\cdot c_{12}\cdot c_{6}$
49	$D+TDP\to^{2k_{3}^{\prime }}TD+PD$	$R_{49}={2k}_{3}^{\prime }\cdot c_{12}\cdot c_{4}$
50	$TE+TE\to^{{2k}_{2}^{\prime }}TET+E$	$R_{50}={2k}_{2}^{\prime }\cdot c_{7}\cdot c_{7}$
51	$TE+TD\to^{k_{2}^{\prime }}TET+D$	$R_{51}={2k}_{2}^{\prime }\cdot c_{7}\cdot c_{8}$
52	$TE+PE\to^{{2k}_{2}^{\prime }}TEP+E$	$R_{52}={2k}_{2}^{\prime }\cdot c_{7}\cdot c_{9}$
53	$TE+PD\to^{k_{2}^{\prime }}TEP+D$	$R_{53}={2k}_{2}^{\prime }\cdot c_{7}\cdot c_{10}$
54	$TD+TE\to^{k_{2}^{\prime }}TDT+E$	$R_{54}={2k}_{2}^{\prime }\cdot c_{8}\cdot c_{7}$
55	$TD+TD\to^{{2k}_{2}^{\prime }}TDT+D$	$R_{55}={2k}_{2}^{\prime }\cdot c_{8}\cdot c_{8}$
56	$TD+PE\to^{k_{2}^{\prime }}TDP+E$	$R_{56}={2k}_{2}^{\prime }\cdot c_{8}\cdot c_{9}$
57	$TD+PD\to^{{2k}_{2}^{\prime }}TDP+D$	$R_{57}={2k}_{2}^{\prime }\cdot c_{8}\cdot c_{10}$
58	$PE+TD\to^{k_{2}^{\prime }}TEP+D$	$R_{58}=k_{2}^{\prime }\cdot c_{9}\cdot c_{8}$
59	$PE+PE\to^{{2k}_{2}^{\prime }}PEP+E$	$R_{59}={2k}_{2}^{\prime }\cdot c_{9}\cdot c_{9}$
60	$PE+PD\to^{k_{2}^{\prime }}PEP+D$	$R_{60}=k_{2}^{\prime }\cdot c_{9}\cdot c_{10}$
61	$PD+TE\to^{k_{2}^{\prime }}TDP+E$	$R_{61}=k_{2}^{\prime }\cdot c_{10}\cdot c_{7}$
62	$PD+PE\to^{k_{2}^{\prime }}PDP+E$	$R_{62}=k_{2}^{\prime }\cdot c_{10}\cdot c_{9}$
63	$PD+PD\to^{{2k}_{2}^{\prime }}PDP+P$	$R_{63}={2k}_{2}^{\prime }\cdot c_{10}\cdot c_{10}$
64	$TE+D\to^{k_{3}^{\prime }}TD+E$	$R_{64}=k_{3}^{\prime }\cdot c_{7}\cdot c_{12}$
65	$TD+E\to^{k_{3}^{\prime }}TE+D$	$R_{65}=k_{3}^{\prime }\cdot c_{8}\cdot c_{11}$
66	$PE+D\to^{k_{3}^{\prime }}PD+E$	$R_{66}=k_{3}^{\prime }\cdot c_{9}\cdot c_{12}$
67	$PD+E\to^{k_{3}^{\prime }}PE+D$	$R_{67}=k_{3}^{\prime }\cdot c_{10}\cdot c_{11}$

**Table 5 polymers-17-00992-t005:** Initial concentrations for the simulation of PET/ODEET interchain exchange in PET:ODET mass ratios of 75:25 (ODEET-75:25), 50:50 (ODEET-50:50) and 25:75 (ODEET-25:75).

Sample Code	Concentration, g/L											
	c_1_	c_2_	c_3_	c_4_	c_5_	c_6_	c_7_	c_8_	c_9_	c_10_	c_11_	c_12_
ODEET-75:25	5.12	0	0	0	0	1.17	0.08	0	0	1.07	0	0
ODEET-50:50	3.41	0	0	0	0	2.34	0.05	0	0	2.14	0	0
ODEET-25:75	1.71	0	0	0	0	3.52	0.03	0	0	3.21	0	0

**Table 6 polymers-17-00992-t006:** Initial concentrations for the simulation of PET/oligoester interchain exchange, where OPEPT, OEPT, OPET, ODEET were used as oligoester; the mass ratio of PET/oligoester is 50:50.

Sample Code	Concentration, g/L											
	c_1_	c_2_	c_3_	c_4_	c_5_	c_6_	c_7_	c_8_	c_9_	c_10_	c_11_	c_12_
OPEPT	3.41	0	0	0	0	2.22	0.05	0	0	2.38	0	0
OEPT	3.41	0	0	0	0	2.29	0.05	0	0	2.23	0	0
OPET	3.41	0	0	0	0	2.10	0.05	0	0	2.63	0	0
ODEET	3.41	0	0	0	0	2.34	0.05	0	0	2.14	0	0

**Table 7 polymers-17-00992-t007:** Molecular weight characteristics of PET/oligoester interchain exchange products, where OPEPT, OEPT, OPET, and ODEET were used as oligoester; the mass ratio of PET/oligoester was 50:50, and the reaction time was 15, 30, 60, 90 min.

Sample Code			M_n_	M_w_	PDI
OPEPT	15	Experimental data	1170	2820	2.41
	30		1520	3430	2.26
	60		1210	2600	2.15
	90		1200	2540	2.12
	Simulation result		1493	2630	1.76
OEPT	15	Experimental data	1450	4990	3.44
	30		1250	5220	4.18
	60		1520	5700	3.75
	90		1700	7590	4.46
	Simulation result		1567	2778	1.77
ODEET	15	Experimental data	1750	3550	2.03
	30		1800	4000	2.22
	60		2000	4900	2.45
	90		2600	5000	1.92
	Simulation result		1621	2883	1.78
OPET	15	Experimental data	1020	2180	2.14
	30		1780	3250	1.83
	60		1270	4010	3.16
	90		1752	4911	2.80
	Simulation result		1383	2410	1.74

**Table 8 polymers-17-00992-t008:** Initial concentrations for the simulation of PET/PEN interchain exchange in PET:ODEET mass ratios of 75:25 (PET/PEN-75:25), 50:50 (PET/PEN-50:50), and 25:75 (PET/PEN-25:75) [21,24].

Sample Code	Concentration, g/L											
	c_1_	c_2_	c_3_	c_4_	c_5_	c_6_	c_7_	c_8_	c_9_	c_10_	c_11_	c_12_
PET/PEN-75:25	5.19	0	0	0	0	1.73	0.05	0	0	0.03	0	0
PET/PEN-50:50	3.46	0	0	0	0	3.46	0.04	0	0	0.06	0	0
PET/PEN-25:75	1.73	0	0	0	0	5.19	0.02	0	0	0.09	0	0

## Data Availability

The original contributions presented in the study are included in the article/Appendix A, further inquiries can be directed to the corresponding authors.

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
