# Peer review of "Modeling the Kinetics of Polyethylene Terephthalate and Polyesters with Terminal Hydroxyl Groups Transesterification Reactions"

_polymers, 2025, doi:10.3390/polym17070992_

Round 1
Reviewer 1 Report
Comments and Suggestions for Authors
In this manuscript, the authors established a kinetic model for the end hydroxy ester exchange reaction between PET and polyester. The models describe the concentration changes of the binding and terminal units of ethylene glycol from PET and ethylene glycol from another polyester during the ester exchange reaction, as well as the concentration changes of free molecules of ethylene glycol and another ethylene glycol. The results of the obtained dynamic model agree well with the experimental results. Research work can contribute to related studies in this field. The research work is designed appropriately, with enough Figures and Tables, and the results can support the conclusions drawn. However, the manuscript has the following issues that need to be strengthened and revised.
(1) What are the assumptions and basis for the established kinetic model? What is the reliability of this assumption and basis? These have yet to be improved.
(2) The methods and equipment section of the manuscript has been written extensively, but it is not clear what each method and equipment specifically measures and how they are linked to the core content of the research. These should be clearly stated, and specific conditions should also be provided.
(3) Please carefully check all formulas and chemical reaction equations to ensure they are accurate.
(4) Almost all Figures do not have vertical coordinates and units, please add them.
(5) Figure 1 and Table 1 are from other literature and are non-essential content that can be deleted.
(6) please give a detailed analysis of all Figures and Tables around the research focus of this manuscript.
(7) The manuscript needs to be carefully read to avoid typos. Such as Eq. (19) in the manuscript is Ea2 instead of Ea1, then 7,80 · 105 should be 7.80 × 105, cm-1 should be cm-1, etc.
Comments on the Quality of English LanguagePlease carefully check all formulas and chemical reaction equations to ensure they are accurate.The manuscript needs to be carefully read to avoid typos. Such as Eq. (19) in the manuscript is Ea2 instead of Ea1, then 7,80 · 105 should be 7.80 × 105, cm-1 should be cm-1, etc.
Author Response
We would like to express our sincere gratitude to the reviewer for the helpful comments and suggestions. We have carefully revisited the article in accordance with all the comments. All the noticeable corrections are marked in red in the text. The responses and comments are below:
Comment 1: What are the assumptions and basis for the established kinetic model? What is the reliability of this assumption and basis? These have yet to be improved.
Responce 1: Thank you for the suggestion. The main assumptions are added in section 3.3. Some of them are described in more detail in the previous article for the kinetic model of the interaction of PET and oligoethylene terephthalates [31] (see doi: 10.3390/polym15143146).
Comment 2: The methods and equipment section of the manuscript has been written extensively, but it is not clear what each method and equipment specifically measures and how they are linked to the core content of the research. These should be clearly stated, and specific conditions should also be provided.
Responce 2: The description of the methods has been shortened and the parameters determined by the research method have been added.
Comment 3. Please carefully check all formulas and chemical reaction equations to ensure they are accurate.
Responce 3: The accuracy of the formulas and chemical reactions was double-checked by all co-authors. Errors found in Equations 19, 20 were corrected.
Comment 4: Almost all Figures do not have vertical coordinates and units, please add them.
Responce 4: For spectra and GPC curves, the absence of vertical axes is common, since the units of measurement in this case do not carry additional information. Therefore, we would prefer not to overload the visual part of the article with extra designations.
Comment 5: Figure 1 and Table 1 are from other literature and are non-essential content that can be deleted.
Responce 5: Figure 1 and Table 1 have been removed. The numbering of figures and tables has been adjusted.
Comment 6: Please give a detailed analysis of all Figures and Tables around the research focus of this manuscript.
Responce 6: According to the new numbering after deleting the previous Figure 1 and Table 1:
Figure 1 – Confirmation of the composition of the commercial form of PET waste, indirect confirmation of the molecular weight required to calculate the concentrations c1 and c7 in the simulation.
Figure 2 – Confirmation of the composition of oligomeric modifiers and their molecular weight distribution. Based on it, the characteristics in Table 1 required to calculate the concentrations c6 and c 0 are determined.
Tables 2, 3, 4 – used designations.
Figures 3, 4 – NMR spectra used further in the verification of the model. The numerical values ​​are given in Table S1 in the supplementary materials.
Tables 5, 6, 8 – input data for the model determined for the experimental conditions. Can be introduced into the calculation file provided in the supplementary materials or used in alternative calculations.
Figure 6 - Confirmation of the composition of oligomeric modifiers and their molecular weight distribution. On its basis, the characteristics in Table 7, necessary for verification, are determined.
Table 7 and Figures 5, 7, 8 - demonstration of the verification of the model, its convergence with experimental data.
Additional comments are also included in the text of the article.
Comment 7: The manuscript needs to be carefully read to avoid typos. Such as Eq. (19) in the manuscript is Ea2 instead of Ea1, then 7,80 · 105 should be 7.80 × 105, cm-1 should be cm-1, etc.
Responce 7: Typos have been corrected, thank you very much for pointing them out.
Reviewer 2 Report
Comments and Suggestions for Authors
The manuscript submitted by Kirshanov et al. (polymers-3513228-peer-review-v1) seems to be interesting. Herein the authors describe a kinetic model and related experimental studies on the change in the concentrations of bound and terminal units of ethylene glycol from PET and glycol from another polyester, as well as free molecules of ethylene glycol and another glycol, during transesterification reactions. The manuscript is well written in terms of the developed kinetic model was found to be in agreement with the experimental data. However, several typos and grammar require improvement. For example,
(1) Please write 1H NMR instead of 1H NMR throughout the manuscript. Also, 1715 cm-1 (not 1715 cm-1, line 226).
(2) Keywords should be improved.
(3) Please check the equations 7, 19, 20, etc.
(4) Please recheck Molecular Weight Characteristics Model Calculations (section 3.5, Figure 4).
(5) Conclusions should be shortened.
Comments on the Quality of English LanguageThe English could be improved to more clearly express the research.
Author Response
We would like to express our sincere gratitude to the reviewer for the helpful comments and suggestions. We have carefully revisited the article in accordance with all the comments. All the noticeable corrections are marked in red in the text. The responses and comments are below:
Comment 1: Please write 1H NMR instead of 1H NMR throughout the manuscript. Also, 1715 cm-1 (not 1715 cm-1, line 226).
Response 1: Typos have been corrected, thank you very much for pointing them out.
Comment 2: Keywords should be improved.
Response 2: The keywords have been adjusted to be more common, and after the adjustment the environmental aspect of the study has been given greater emphasis.
Comment 3: Please check the equations 7, 19, 20, etc.
Response 3: Thank you for the suggestion. Equation 7 has been removed, and errors found in Equations 19, 20 have been corrected.
Comment 4: Please recheck Molecular Weight Characteristics Model Calculations (section 3.5, Figure 4).
Response 4: The accuracy of the formulas and chemical reactions was double-checked by all co-authors.
Comment 5: Conclusions should be shortened.
Response 5: The conclusion has been shortened to be more concise.
Reviewer 3 Report
Comments and Suggestions for Authors
The manuscript entitled “Modeling the Kinetics of PET and Polyesters with Terminal Hydroxyl Groups Transesterification Reactions” is well written and I will consider it for publication in Polymers after minor revision as follow:
- Please highlight novelty of this project in the abstract section.
- The authors have calculated vibrational frequency of FTIR. It would be interesting to calculate frequency scaling factors for vibrational assignment. (See: doi: 10.1016/j.diamond.2022.109142).
- Abbreviations should be defined at first mention and used consistently thereafter. (Such as oligo (ethylene phthalate-co-terephthalate) (OEPT), oligo (propyl-133 ene-co-ethylene terephthalate) (OPET) and oligo (propylene-co-ethylene phthalate-co-ter-134 ephthalate) (OPEPT) and many more).
Overall, I think that the work presented in the current manuscript merit publication in the journal after above mentioned improvement.
Author Response
We would like to express our sincere gratitude to the reviewer for the helpful comments and suggestions. We have carefully revisited the article in accordance with all the comments. All the noticeable corrections are marked in red in the text. The responses and comments are below:
Comment 1: Please highlight novelty of this project in the abstract section.
Response 1: The novelty was further emphasized in the abstract.
Comment 2: The authors have calculated vibrational frequency of FTIR. It would be interesting to calculate frequency scaling factors for vibrational assignment. (See: doi: 10.1016/j.diamond.2022.109142).
Response 2: Thank you for your recommendation, we will study the method in more detail and intend to apply it in future research on this topic.
Comment 3: Abbreviations should be defined at first mention and used consistently thereafter.
Response 3: The necessary adjustments have been made. Abbreviations are defined when first used, as well as in the Abbreviations section at the end of the article. Abbreviations are used extensively later in the paper, primarily in figures and tables.
Reviewer 4 Report
Comments and Suggestions for Authors
The article polymers-3513228 entitled "Modeling the Kinetics of PET and Polyesters with Terminal 2Hydroxyl Groups Transesterification Reactions" has been reviewed. My review report is given below. In the present form it cannot be recommended for acceptance provided the authors are ready to modify their article in accordance with the below mentioned issues.
This research presents a significant advancement in the field of polyethylene terephthalate (PET) chemical recycling by utilizing an interchain exchange mechanism via transesterification. The study proposes a kinetic model that describes the changes in various molecular species' concentrations during the reaction, offering a quantitative understanding of the process. My opinion is listed as below.
1. The study should discuss in detail any simplifying assumptions in the kinetic model, such as ideal mixing, reaction equilibrium, or steric effects that might influence transesterification.
2. A comparison with classical chemical recycling methods would strengthen the claim that this approach effectively bypasses molecular weight reduction.
3. The observed discrepancies between simulated and experimental weight-average molecular weights (Mw) and polydispersity indices (PDI) suggest that some assumptions in the model may oversimplify reaction dynamics.
4. The study does not mention whether catalysts were used to enhance the transesterification reaction. Investigating the impact of different catalysts could provide further insights.
5. The study focuses on 280°C but does not deeply explore temperature-dependent kinetic variations beyond comparisons with PET/PEN at 280°C and 300°C.
6. While the kinetic model is promising, its scalability to industrial processes and potential economic feasibility remain unaddressed.
7. How does the proposed kinetic model account for reaction reversibility in the transesterification process? Does the model consider steric hindrance effects on the interchain exchange rates? What role do catalysts or reaction conditions (temperature, pressure) play in the reaction kinetics?
8. How do molecular weight distributions of the products compare to those obtained via traditional PET chemical recycling? Are there any by-products formed during the interchain exchange, and how do they affect the final polymer properties?
9. What are the major similarities and differences between the interchain exchange kinetics of PET/oligodiethylene terephthalate and PET/PEN blends? How does the presence of aromatic and aliphatic components in the oligomer influence reaction selectivity?
10. Does this method improve the recyclability of PET in a more energy-efficient manner compared to classical depolymerization methods? What are the practical implications for polymer recycling industries, and how does the process compare in terms of sustainability?
11. The study mainly focuses on PET and oligodiethylene glycol terephthalate. It would be valuable to see how the model applies to other copolyesters with different glycol and acid components (e.g., polybutylene terephthalate, polytrimethylene terephthalate).
12. How does the proposed method compare to enzymatic or solvent-based PET recycling techniques in terms of efficiency and molecular weight retention?
Future research should address temperature-dependent kinetics, catalyst effects, and steric limitations to enhance predictive accuracy. Additionally, scalability and real-world applicability should be explored to maximize the impact of this work in sustainable polymer processing.
The authors need to focus on these points and expected to incorporate suitable changes by considering the above issues.
Author Response
We would like to express our sincere gratitude to the reviewer for the helpful comments and suggestions. We have carefully revisited the article in accordance with all the comments. All the noticeable corrections are marked in red in the text. The responses and comments are below:
Comment 1: The study should discuss in detail any simplifying assumptions in the kinetic model, such as ideal mixing, reaction equilibrium, or steric effects that might influence transesterification.
Response 1: Thank you for the suggestion. The main assumptions are added in section 3.3. Some of them are described in more detail in the previous article for the kinetic model of the interaction of PET and oligoethylene terephthalates [31] (see doi: 10.3390/polym15143146).
Comment 2: A comparison with classical chemical recycling methods would strengthen the claim that this approach effectively bypasses molecular weight reduction.
Response 2: We would like to avoid further increasing the size of the already lengthy Introduction. But your comment is really important, so we have included a short explanation with a link to a previous work that explains it in more detail [7, 14] (see doi: 10.3390/polym14081602, Figure 1 and its legend; 10.3390/polym14040684).
Comment 3: The observed discrepancies between simulated and experimental weight-average molecular weights (Mw) and polydispersity indices (PDI) suggest that some assumptions in the model may oversimplify reaction dynamics.
Response 3: Indeed, using the ideal Flory-Schulz distribution does not allow us to predict the weight-average molecular mass with high accuracy. In further development of the model, we plan to explore the use of other simplifications.
Comment 4: The study does not mention whether catalysts were used to enhance the transesterification reaction. Investigating the impact of different catalysts could provide further insights.
Response 4: Thank you for this extremely valuable comment! The missing information about the catalyst has been added to section 2.3, 3.3. Investigation of various catalysts and solving the inverse kinetic problem for various catalysts using our model is an important task for our future research.
Comment 5: The study focuses on 280°C but does not deeply explore temperature-dependent kinetic variations beyond comparisons with PET/PEN at 280°C and 300°C.
Response 5: In the described PET/PEN blend, both polymers have a high molecular weight, so at a low concentration of end groups, the processes of interchain exchange are slow, and we can easily see the time dependence and verify the kinetics. When studying the PET/PDET blend at high temperature, we see a very rapid achievement of equilibrium concentrations (see the attached file), so we did not include these data in the text of the article.
Comment 6: While the kinetic model is promising, its scalability to industrial processes and potential economic feasibility remain unaddressed.
Response 6: The creation of a pilot plant and scaling up the process, accompanied by verification of the model, is our plan for further research. We assume that the model may have high potential in terms of optimizing recipes and selecting optimal conditions.
Comment 7: How does the proposed kinetic model account for reaction reversibility in the transesterification process? Does the model consider steric hindrance effects on the interchain exchange rates? What role do catalysts or reaction conditions (temperature, pressure) play in the reaction kinetics?
Response 7: All forward and reverse reactions occurring during transesterification are included as separate reactions in Table 4. Comments on chain and catalyst effects are included in Section 3.3. The effect of temperature is taken into account via the Arenius equation. We assume that pressure will not play a significant role for the process in the liquid phase, but the study of different types of catalysts, including heterophase ones, and the effect of pressure on them is an urgent future task.
Comment 8: How do molecular weight distributions of the products compare to those obtained via traditional PET chemical recycling? Are there any by-products formed during the interchain exchange, and how do they affect the final polymer properties?
Response 8: Average molecular weights directly depend on the molecular weight of the agent used in the recycling method. As for the width of the molecular weight distribution, we previously found that the polydispersity index is higher in the interchain exchange reaction with oligoesters than in classical glycolysis [7, 14]. As noted earlier, studying and taking this effect into account is one of the further tasks. At present, the investigation of side reactions was not the objective of our study. In the future, we plan to include them in the model, primarily the etherification reaction.
Comment 9: What are the major similarities and differences between the interchain exchange kinetics of PET/oligodiethylene terephthalate and PET/PEN blends? How does the presence of aromatic and aliphatic components in the oligomer influence reaction selectivity?
Response 9: PET, PEN and oligodiethylene terephthalate (ODET) have similar structures, so the main differences we found are related to the reaction rate due to different molecular weights and, consequently, different concentrations of the most active end units.
Comment 10: Does this method improve the recyclability of PET in a more energy-efficient manner compared to classical depolymerization methods? What are the practical implications for polymer recycling industries, and how does the process compare in terms of sustainability?
Response 10: From the point of view of energy efficiency, all transesterification reactions are quite sustainable. However, in classical heterogeneous glycolysis, excess glycol is used, the removal and purification of which require significant energy costs. In interchain exchange processes, recycling agents are completely incorporated into the product and do not require separation.
Comment 11: The study mainly focuses on PET and oligodiethylene glycol terephthalate. It would be valuable to see how the model applies to other copolyesters with different glycol and acid components (e.g., polybutylene terephthalate, polytrimethylene terephthalate).
Response 11: Indeed, in the future, we plan to study various blends, including such a common industrial polymer as PBT, as well as aliphatic polyesters based on adipate.
Comment 12: How does the proposed method compare to enzymatic or solvent-based PET recycling techniques in terms of efficiency and molecular weight retention?
Response 12: In the previous work [31] (see doi: 10.3390/polym15143146), a similar model for the interaction of PET and oligoethylene terephthalates, bis(2-hydroxyethyl) terephthalate and ethylene glycol equally successfully described various homogeneous processes – both interchain exchange and glycolysis with ethylene glycol in solution.
On behalf of the team of authors, I thank you for your valuable and detailed comments. We have taken your recommendations into account in this work and plan to use them in further research.

Round 2
Reviewer 1 Report
Comments and Suggestions for Authors
I would like to thank the authors for their efforts to revise the manuscript. But I still insist that the vertical coordinates and units should be added in the Figures since it may mislead the readers. For example, in FTIR, vertical coordinates can be expressed in at least two ways: absorbance and transmittance. The results of the two expression methods are completely opposite. Therefore, the annotation of the vertical axis and its units is very important.
Author Response
We are grateful to the reviewer for further clarification of his comment. Indeed, in the case of FTIR and many others, it is necessary to clearly distinguish the vertical coordinates used.
Comment 1: The vertical coordinates and units should be added in the Figures since it may mislead the readers.
Responce 1: The missing vertical axes and units annotations have been added in Figures 1, 2, 3, 6.
Reviewer 4 Report
Comments and Suggestions for Authors
The article polymers-3513228-V2 has been reviewed. After going through the manuscript, it is learnt that, the authors have undertaken all issues and corrected them and further incorporated them in the manuscript accordingly. Hence, at this moment I am of the opinion that, the manuscript may e accepted in the present form without any further modification. I recommend that the article may now be accepted and published in the esteemed journal.
Author Response
We are grateful to the reviewer for his careful consideration of our article. The comments provided allowed us to significantly improve this work and plan further research on the topic of polyethylene terephthalate chemical recycling.
Round 3
Reviewer 1 Report
Comments and Suggestions for Authors
It can be accepted for publication as it is.